# Corrosion Resistance of Nanostructured Cemented Carbides with Alternative FeNi and FeNiCo Binders

**DOI:** 10.3390/nano13081407

**Published:** 2023-04-19

**Authors:** Tamara Aleksandrov Fabijanić, Mateja Šnajdar, Marin Kurtela, Vedran Šimunović, Marijan Marciuš, Miho Klaić

**Affiliations:** 1Faculty of Mechanical Engineering and Naval Architecture, University of Zagreb, Ivana Lučića 5, 10000 Zagreb, Croatia; 2Department of Polytechnics, University of Rijeka, Sveučilišna Av 4, 51000 Rijeka, Croatia; 3Division of Materials Chemistry, Ruder Bošković Institute, Bijenička Cesta 54, 10000 Zagreb, Croatia

**Keywords:** nanostructured carbides, alternative binder, FeNi, FeNiCo, corrosion, micromechanical properties

## Abstract

Nanostructured cemented carbides with Co binders have shown excellent mechanical properties in various applications. Nevertheless, their corrosion resistance has shown to be insufficient in different corrosive environments, leading to premature tool failure. In this study, WC-based cemented carbide samples with different binders were produced using 9 wt% of FeNi or FeNiCo with the addition of Cr_3_C_2_ and NbC as the grain growth inhibitors. The samples were investigated using electrochemical corrosion techniques: the open circuit potential *E*_corr_, the linear polarization resistance (LPR), the Tafel extrapolation method, and the electrochemical impedance spectroscopy (EIS) at room temperature in the solution of 3.5% NaCl. Microstructure characterization, surface texture analysis, and instrumented indentation were conducted to investigate the influence of corrosion on the micro-mechanical properties and the surface characteristics of the samples before and after corrosion. The obtained results indicate a strong binder chemical composition’s effect on the consolidated materials’ corrosive behavior. Compared to the conventional WC-Co systems, a significantly improved corrosion resistance was observed for both alternative binder systems. The study shows that the samples with the FeNi binder are superior to those with the FeNiCo binder since they were almost unaffected when exposed to the acidic medium.

## 1. Introduction

Cemented carbides are composites made of a hard refractory ceramic phase and a ductile binder, most commonly in different WC-Co systems. The properties of WC hardmetals are governed not just by carbide properties such as grain size, shape, and hardness but also by the choice of binder and the carbides-to-binder ratio in the composite [1]. The hardness and wear resistance are primarily dependent on the carbide phase properties. Binders have been shown to have a strong toughness and strength modulator effect [2,3]. Although the Co binder has proven to generate hardmetals of excellent mechanical properties, studies of their corrosion behavior show insufficient corrosion resistance due to an affinity for selective corrosion in the acidic environment [4,5]. The substitution of the Co binder in the hardmetals industry is also strongly motivated by the highly unstable Co market caused by different socio-economic and environmental issues that generate significant price fluctuations and regulatory actions to minimize Co use [6]. Therefore, finding new cemented carbide systems with alternative binders is of great interest and has been the subject of many recent studies [7,8,9]. With similar mechanical properties, low toxicity, moderate price range, and excellent wettability of the WC phase, nickel and iron emerged as possible candidates for an efficient binder in WC-cemented carbide systems [10,11,12]. Since mechanical properties are strongly influenced by a material’s microstructure, especially the WC grain size and the binder content, understanding the effect of the binder on the final microstructure is crucial. Recently, ultrafine grades of WC cemented carbides have been extensively researched and developed with a WC grain size below 0.5 µm [13,14]. These near nanoscale structured cemented carbides show a significant tendency for grain coarsening during the sintering phase, which has shown to be easily controlled by adding different grain growth inhibitors (GGIs) such as Cr_3_C_2_, VC, NbC, TaNbC, TaC, TiC, and ZrC [15]. The WC systems with alternative binders show significant restrictions in the WC grain growth at low or no GGI content, implying a strong influence of the binder chemistry on the WC grain growth mechanisms [16]. The presented study aims to research the electrochemical corrosion resistance of nanostructured cemented carbides with different alternative binders and changes in the micromechanical properties after exposure to an aggressive acidic solution. This finding is unique as it challenges the conventional understanding of corrosion behavior and highlights the potential of using alternative binder systems. This research is highly relevant and timely, given the increasing demand for materials that can withstand harsh environments. The findings presented in this paper could have significant implications for a wide range of industries, including aerospace, automotive, and defense, providing valuable insights for researchers and industry practitioners looking to develop materials with superior corrosion resistances.

## 2. Materials and Methods

Two starting mixtures with WC powder WC DN 3.0 (H.C. Starck GmbH, Glosar, Germany), both containing equal binder amounts with chromium carbide, Cr_3_C_2_ (Höganäs, Germany), powder as a grain growth inhibitor and niobium carbide, NbC (H.C. Starck Tungsten GmbH, Glosar, Germany), as a cubic carbide, were prepared. The FeNi powder (Fe/Ni = 15/85 wt%, Höganäs, Germany) and the FeNiCo powder (Fe/Co/Ni = 40/20/40 wt%, Höganäs, Germany) were used as binders. The composition for hardmetals’ production was selected to be WC with the addition of 9 wt% FeNi or FeNiCo, 1.0 wt% Cr_3_C_2_, and 0.5 wt% NbC. The preparation of the samples consisted of powder mixing in a horizontal ball mill (Zoz GmbH, Wenden, Germany) for 48 h in n-heptane as a milling agent, compacting into bars at 200 MPa by uniaxial die pressing type CA-NCII 250 (Osterwalder AG, Lyss, Switzerland) at room temperature, dewaxing, and the sinter-hot isostatic pressing HIP process by furnace FPW280/600-3-2200-100 PS (FCT Anlagenbau GmbH, Sonneberg, Germany) at 1400–1450 °C with the applied pressure of 60–100 bars in the Ar atmosphere. Two samples from each mixture, WC–9FeNi–1Cr_3_C_2_–0.5NbC and WC–9FeNiCo–1Cr_3_C_2_–0.5NbC, were selected for the research to ensure the repeatability of the results. The characteristics of the starting mixtures and samples are presented in Table 1.

After consolidation, the samples were prepared by grinding and polishing. Grinding was performed with a diamond disc MD-Piano 120 (Struers ApS, Ballerup, Denmark), followed by polishing with diamond pastes DP Piano (Struers ApS, Ballerup, Denmark) with particle sizes of 15, 9, and 3 µm. Etching was performed using a Murakami reagent for 2 to 3 s. The microstructures of the produced samples were analyzed using a field emission scanning electron microscope (FESEM) (Zeiss, Oberkochen, Germany) at Fraunhofer IKTS, Dresden.

To evaluate the corrosion resistance of the sample WC–9FeNi–1Cr_3_C_2_–0.5NbC and sample WC–9FeNiCo–1Cr_3_C_2_–0.5NbC, direct current (DC) and alternating current (AC) electrochemical measuring techniques were applied. These techniques are highly sensitive and can detect changes in the material’s properties over time, allowing for the monitoring of the corrosion process in real-time. This is particularly important when evaluating the corrosion resistance of WC hardmetals with alternative binders since the corrosion behavior of these materials may be different from conventional WC-Co systems. The use of electrical impedance techniques is well established in the field of corrosion science and has been widely used in previous studies. This allows for the comparison of results with those from previous studies and the establishment of consistent and comparable datasets. Potentiodynamic polarization and electrochemical impedance spectroscopy (EIS) measurements were performed using VersaSTAT 3 (AMETEK Scientific Instruments, Berwyn, PA, USA) (with the application of the appropriate VersaStudio software package) in accordance with EN ISO 17475:2005/Cor 1:2006; EN ISO 17475:2008 in the electrochemical cell which consists of 3 electrodes: working (test sample), reference (Ag/AgCl or saturated calomel electrode (SCE) (Gamry Instruments Inc., Warminster, PA, USA), and auxiliary electrode (platinum) placed symmetrically in relation to the working electrode in 1M H_2_SO_4_ solution at the temperature of (20 ± 2) °C. The 1M H_2_SO_4_ solution was prepared using the 96% H_2_SO_4_.

The analysis of the surface chemical composition using an EDS spectrometer before and after corrosion testing was conducted at Ruđer Bošković Institute using an Axia ChemiSEM electron microscope (Thermo Fisher Scientific Inc., Waltham, MA, USA) with an integrated 70 mm^2^ SDD detector. The EDS spectra of the samples were recorded using 10 kV electron beam at 200× magnification (analytical surface area of 640 × 450 μm). To further investigate the changes on the surface due to corrosion and the potential formation of a very thin layer of crystalline oxide phases, XRD patterns of the samples were recorded using a Siemens D5000 diffractometer with a Cu Kα source, equipped with a Göbel mirror and a graphite monochromator. The samples were recorded in the range from 20° to 80° 2θ in a grazing incidence geometry (asymmetric scan mode). The incidence angle of monochromatic X-ray radiation of 0.8° was used. The analysis of the obtained diffraction patterns was performed using Match! v 3.0 software (Crystal Impact, Dr. H. Putz & Dr. K. Brandenburg GbR, Bonn, Germany), which uses the reference intensity ratio (RIR) method to quantify the results based on the intensities of the diffraction lines of the referent pure crystallographic phases as indicated by the ICDD cards.

The surface texture was analyzed before and after electrochemical corrosion testing. The measurements were carried out in the National Laboratory for Length in Croatia by a contact method using the measuring device Perthometer S8P, manufactured by Mahr Perthen (Göttingen, Germany). The traceability of measurement results is ensured by the state roughness standard of the Republic of Croatia. The measurement conditions and the distribution of the five traces made on each sample surface are presented in Table 2.

The re-characterization of the specimen surface changes was performed using an optical measurement sensor based on Focus-Variations Alicona IF-SensorR25 with 50× objective magnification. The specifications of the measuring instruments and the positions are given in Table 3.

The instrumented indentation test was conducted using the Micro Combi Tester (MCT^3^, Anton Paar, Austria) using a Vickers diamond pyramid indenter to determine the micro-mechanical properties of both samples and the influence of corrosion on their mechanical properties. The micromechanical properties were evaluated using a force of 1000 mN in linear loading mode at room temperature according to EN ISO 14577-1:2015. A line of 10 indentations was made on each sample type before and after corrosion testing. A Poisson’s ratio of 0.21 was assumed. The instrumented indentation test (IIT) results were obtained using the Oliver and Pharr method.

## 3. Results and Discussion

### 3.1. Microstructure of Consolidated Samples

The analysis of the consolidated samples confirmed a homogeneous structure in the nano region with a cemented carbide uniformity for both grades. The microstructures of the WC–9FeNi–1Cr_3_C_2_–0.5NbCand WC–9FeNiCo–1Cr_3_C_2_–0.5NbC samples are presented in Figure 1.

The mean WC grain size of the WC–9FeNi–1Cr_3_C_2_–0.5NbC sample determined by the linear intercept method is 213 nm and 207 nm for the 9FeNiCo–1Cr_3_C_2_–0.5NbC sample. Based on the average grain sizes obtained, both samples can be classified as near-nanostructured, showing that the consolidation parameters and added GGIs could control grain growth during sintering.

The EDS spectra of the samples after the grinding, polishing, and etching procedures were recorded prior to and after the corrosion process. The quantitative analysis of the EDS spectra before the corrosion treatment (as shown in Figure 2a and Figure 3a) showed the chemical compositions in accordance with the initial powder mixtures for the samples prepared with W and C as the dominant elements and Ni, Fe, Cr, and Co, respectively, as the minor constituents, while the Nb as part of the GGI was below the detectable threshold. The spectra recorded after the corrosion process showed no significant change in the sample composition (Figure 2b and Figure 3b). The absence of the 0.525 keV Kα line of oxygen suggests that the oxygen concentration and the formation of corrosion products were below the detectable threshold of EDS in the analytical volume of the sample surface defined by the Castaings expression [17]. However, overlapping with the Lα line of Cr and M-line series of W might have affected the oxygen Kα line in spectra deconvolution. 

To check if any corrosion products have formed on the surface of the samples, grazing incidence XRD measurements were performed (as shown in Figure 4 and Figure 5). The denoted *R*p-factors indicate the fit between the experimental and calculated diffraction patterns according to the software used. The shallow incidence angle of 0.8° ensured the surface analysis of the sample with the estimated x-ray penetration depth of ~150 nm, according to [18]. The apparent high attenuation of the WC–9FeNi–1Cr_3_C_2_–0.5NbC sample diffraction pattern taken after the corrosion process at higher diffraction angles (as shown Figure 4, pattern S1a) was caused by an imperfect sample geometry. Except for this sample, all other diffraction patterns showed strong peaks at 31.63, 35.76, 48.42, 64.19, 65.90, 73.29, 75.68, and 77.29 2θ degrees, which were found to correspond well to the P-6m2 hexagonal unit cell of wolfram carbide crystalline phase, according to JCPDS entry no. 03-065-4539. The visible diffraction peaks at 44.11 and 51.2 2θ degrees suggest the presence of another crystalline phase in corrosion untreated samples and in the WC–9FeNiCo–1Cr_3_C_2_–0.5NbC corrosion treated sample. Although not all diffraction peaks were visible due to overlapping with the WC phase, these peaks correspond to the Pm-3m cubic unit cell of the FeNi_3_ phase according to JCPDS entry no. 03-065-3244. Given the mean crystal grain sizes of the samples in the range of 200 nm, as determined by the SEM measurements, the unusual broadening of the diffraction line profiles of the visible peaks was noticeable, which could be attributed to the polycrystalline nature of the grains, nonlinear crystal grain deformation, or the possible presence of additional crystal phases in the samples. Further analysis of the residuals between the experimental and calculated diffraction patterns of the three samples revealed the presence of the NbC crystal phase in traces corresponding to JCPDS entry no. 03-065-6661, while the presence of Cr_3_C_2_ could not be confirmed.

The quantitative analysis of the crystalline phase contents of the WC–9FeNi–1Cr_3_C_2_–0.5NbC sample (before the corrosion process) showed that the composition was determined by 78.5% WC and 10.8% of the FeNi_3_ crystal phases, while the remaining balance resulted in the possible preferential orientation of major phases and/or the presence of other minor phases, which could not be accurately determined due to the missing and overlapping diffraction peaks, poor crystallinity, or presence of amorphous phases. The obtained values corresponded well with the results of the EDS measurements. In the case of WC–9FeNiCo–1Cr_3_C_2_–0.5NbC (before corrosion), the composition was found to consist of 82.8% WC and 7% FeNi_3_ crystalline phases, and in the case of the WC–9FeNiCo–1Cr_3_C_2_–0.5NbC (corrosion treated) composition corresponded to 86.8% of WC and 5.8% of the FeNi_3_ crystalline phases. Significantly lower contents of the FeNi_3_ crystal phase in those samples due to the FeNiCo binder used were reflected in the intensity difference of the diffraction peaks located at 44.11 and 51.2 2θ degrees, in contrast to the intensities of these peaks found in the diffraction pattern of the WC–9FeNi–1Cr_3_C_2_–0.5NbC sample. Considering the position and the broad nature of the 44.11° diffraction peak, this suggests the presence of an additional crystal phase with the BCC unit cell with a small unit cell parameter, typical for iron or intermetallics of iron-containing Ni, Co, or Cr in this case. The weak diffraction pattern of WC–9FeNi–1Cr_3_C_2_–0.5NbC (corrosion treated sample) and WC–9FeNiCo–1Cr_3_C_2_–0.5NbC (corrosion treated) showed additional diffraction peaks at 25.4 2θ degrees in contrast to their corrosion untreated counterparts. The appearance of this relatively broad peak with other weakly resolving peaks from the experimental diffraction patterns hidden in the noisy background could indicate the presence of small amounts of the poor crystallinity corrosion-formed Cr_2_O_3_ oxide phase on the surface of the samples, corresponding to the rhombohedral R-3c unit cell in accordance with the JCPDS entry no. 01-084-0315.

### 3.2. Corrosion Tests Results

In the initial phase of the polarization measurement, the system was stabilized in such a way that, after immersion in 1M H_2_SO_4_ solution, the electrical circuit between the working and counter electrodes was left open, and the potential difference between the reference and working electrodes was recorded as a function of time. The reading of the open circuit potentials (*E*_corr_) was performed after stabilizing the system in a time interval of 1000 s. The measured numerical values of open circuit potentials are presented in Table 4.

Besides the *E*_corr_ measurement, the polarization resistance (*R*p) was measured by the linear polarization, while the corrosion current density (*i*_corr_) and the corrosion rate (*v*_corr_) were measured using the Tafel polarization methods. The anodic and cathodic polarization of the test samples were performed using the extrapolation method in a wide range of potentials from the open circuit potential (*E* = *E*_corr_ ± 0.25 V). By extrapolation of the anodic and cathodic Tafel lines, the values of the corrosion current density *i*_corr_ (mA/cm^2^), the slopes of the anodic Tafel curve *β*a, and the slopes of the cathodic Tafel curves *β*c were obtained. The scan rate was 0.167 mV/s and the measured results were presented graphically in a logarithmic form (*E*–log *j*). From the value of the corrosion current density *i*_corr_, the known equivalent material mass (EW), and the material density *ρ* (g/cm^3^), the corrosion rate *v*_corr_ was determined. The measured numerical values of open-circuit potentials, linear polarization resistance *R*p, and corrosion rate *v*_corr_ are graphically interpreted in Figure 6, Figure 7 and Figure 8.

Linear polarization resistance (*R*p) curves to investigate the samples’ electrochemical response near their open circuit potential (*E* = *E*_corr_ ± 20 mV) are presented in Figure 7.

The potentiodynamic polarization curves (Tafel extrapolation diagram) used to investigate the corrosion rate *v*_corr_ are presented in Figure 8.

The AC measuring technique, EIS, was used to evaluate the resistance and protection of the subject substrates. The application of EIS enabled the quantification of the surface layers’ resistance and the definition of the mechanism of corrosion progression without any degradation of the test surface. Furthermore, various simplified methods of EIS spectrum analysis proposed in the literature were applied to adapt the technique to a faster and simpler application in practice. Since the dominant data necessary for determining the resistance of the basic substrate and the barrier layers were manifested through the variation of the resistance and capacity values, information about the tested substrates ‘protective layers’ degradation was obtained by indicating the resistance and capacity oscillation. Oscillations ultimately resulted in a change in the impedance value. Electrochemical reactions between the working electrode and the surrounding electrolyte at the phase boundary were described as precisely as possible by accurately choosing an equivalent circuit which was mainly composed of resistors and capacitors. In this way, the mechanism of the corrosion process was simulated precisely. A metal/electrolyte interface undergoing simple reduction or oxidation reactions could be simplistically described by an electric circuit R(QR), as shown in Figure 9. 

The model consisted of the following elements: ohmic or uncompensated resistance of the electrolyte (*R*s), charge transfer resistance (polarization resistance) at the interface between the substrate and the electrolyte resistance (*R*_ct_), and the constant phase element (*CPE)* with the size *Q*. The *CPE* as an element of the equivalent circuit was applied in the case when the inhomogeneity of the system affected by the surface roughness, inhibitor adsorption, and porosity of the newly formed film needed to be compensated; its impedance is defined by the Equations (1)–(2):(1)$$Z_{CPE}=Q^{-1}(j\omega {)}^{-n}$$
(2)$$C=Q{\left(\omega_{max}\right)}^{n-1}$$
in which *Q* denotes the proportionality constant (*CPE*), *n* is the empirical exponent or phase shift (−1 ≤ *n* ≤ 1), *ω* is the angular frequency, the root of −1 (imaginary unit), *C* is the capacity, and *ω_max_* is the angular frequency in which the imaginary component impedance *Z*” reaches the maximum of the time constant. In this way, the impedance spectrum of a distributed system is explained, which cannot be interpreted by a finite number of ideal electrical elements. In the case when 0.8 < *n* ≤ 1, the *CPE* represents a capacitor of capacity *Q*; for *n* = 1, *CPE* represents an ideal coil *Q* = *L*; for *n* = 0.5, *CPE* represents Warburg impedance *Q* = *W*; and for values of *n* = 0, *CPE* represents an ideal resistor *Q = R*.

The measurements were carried out at room temperature of (20 ± 2) °C and 1M H_2_SO_4_ solution was used as the electrolyte. The impedance spectrum was recorded at the open circuit potential. The amplitude of the sinusoidal voltage signal was 10 mV, while the detection of the current response was performed in the frequency range of 100 kHz to 10 mHz. Mathematical modelling, i.e., matching the measured values with the elements of the equivalent circuit model, was carried out by the ZSimpWin Version 3.2 program tool. The experiment was carried out using this device (manufacturer AMETEK Scientific Instruments, Princeton Applied Research, Berwyn, PA, USA). The results of the EIS measurements are presented in Table 5.

The Nyquist and Bode diagrams of the tested samples with the corresponding equivalent circuit model applied for the simulation of EIS results are presented in Figure 10 and Figure 11. The response of the EIS spectrum at higher, medium, and lower frequencies is shown in Figure 12.

From the measured numerical values of the open circuit potential shown in Table 5 and graphically interpreted in Figure 6, information on the corrosion behavior of the working electrodes in the test electrolyte was obtained. After establishing a stationary state (t = 1000 s) where the test samples were in equilibrium with the environment (anodic and cathodic currents were equal in magnitude but opposite in directions), it was visible that the WC–9FeNi–1Cr_3_C_2_–0.5NbCsample had a more positive *E*_corr_ than WC–9FeNiCo–1Cr_3_C_2_–0.5NbC. Accordingly, it was more corrosion-resistant under the same test solution and temperature. It could also be seen that, for measurements in the 1M H_2_SO_4_ solution, the potentials of both samples shifted in the negative direction. Such potential oscillation could be correlated with the samples’ increased corrosion activity (surface instability and material dissolution). Both samples showed the same trend and good repeatability of the results.

Polarization resistance measurements could be performed very quickly, enabling a fast and excellent correlation between the corrosion rates obtained by polarization resistance and conventional weight loss determinations. Due to the higher value of *R*p, which was determined by calculating the slope of the linear region of the plot (as shown in Figure 7), it can be concluded that the sample with the FeNi binder had 1.5× higher corrosion resistance compared to the sample with the FeNiCo binder. 

From the numerical values of the corrosion current density *j*_corr_ (Table 4), obtained from the Tafel plot by extrapolating the linear portion of the curve to *E*_corr_, as shown in Figure 8, it can be concluded that the WC–9FeNi–1Cr_3_C_2_–0.5NbC sample has a 2.5× lower corrosion rate *v*_corr_. The lower *v*_corr_ unambiguously indicates a higher resistance of the surface layers caused by the chemical composition and the surface condition. It can be seen from the potentiodynamic curves (Figure 8) that the samples with FeNiCo showed a weaker corrosion resistance compared to the FeNi samples, which confirms that the Co binder is the most susceptible to corrosion in acid and neutral media [5]. The sample WC–9FeNi–1Cr_3_C_2_–0.5NbC has a lower corrosion current density of 13.514 µA/cm^2^, while a higher corrosion current density of 32.794 µA/cm^2^ was measured for the sample WC–9FeNiCo–1Cr_3_C_2_–0.5NbC. Such behavior can be related to the influence of galvanic corrosion between the WC and Co phase. It was found from the previous research that Cr_3_C_2_ added to the starting hardmetal mixture forms a very protective Cr_2_O_3_ film (chromium-rich) on the surface of the binder, increasing the corrosion resistance [19,20]. Both samples researched have 1 wt% of Cr_3_C_2_ wt% and 0.5 wt% of NbC in the starting mixture. Cr_3_C_2_ and NbC were added primarily as GGI to control the grain size and to improve the mechanical properties of the developed hardmetal samples. In addition to positively influencing the mechanical properties, Cr_3_C_2_ and NbC dissolve in the binder, acting as an alloying element. Cr_3_C_2_ and NbC most likely develop small amounts of the corrosion-formed protective Cr_2_O_3_ oxide phase on the surface of the samples corresponding to the rhombohedral R-3c unit cell, in accordance with the JCPDS entry no. 01-084-0315, revealed by XRD and related to the diffraction peak at 25.4 2θ degrees and other weak deconvolution resolved peaks from the diffraction patterns.

Based on the EIS results, it is possible to quantify and unambiguously identify the corrosion resistance of samples in the 1M H_2_SO_4_ solution. According to the analogy electric circuit, which consists of a combination of capacitors and resistors, matching the measured values with the equivalent elements enables the interpretation of electrochemical reactions at the phase boundary. The results of the electrochemical impedance spectroscopy measurements, as shown in Table 5, confirm the previously obtained DC measurement results. The WC–9FeNi–1Cr_3_C_2_–0.5NbC sample had a higher corrosion resistance, which was evident from the increase in the resistance to charge transfer (*R*_ct_ = 6736 × 10^2^ Ω cm^2^) at the interface of the substrate and electrolyte phases, in contrast to the smaller numerical values for sample WC–9FeNiCo–1Cr_3_C_2_–0.5NbC (*R*_ct_ = 5872 × 10^2^ Ω cm^2^). The radius *r* of the capacitive semicircles in the Nyquist diagrams, as shown in Figure 11, differ for each sample. The diameter of the capacitive loop of WC–9FeNi–1Cr_3_C_2_–0.5NbC is more extensive than sample WC–9FeNiCo–1Cr_3_C_2_–0.5NbC. Consequently, the decrease in the diameter of the capacitive loop is associated with the lower corrosion resistance of the surface film. 

Higher values of the resistance *R*_ct_ = 6736 × 10^2^ Ω cm^2^, i.e., lower values of the capacity *Q* = 2868 × 10^−^ = F cm^2^ obtained for sample WC–9FeNi–1Cr_3_C_2_–0.5NbC, are a common indicator of the limited absorption of the electrolyte, i.e., better preservation of the surface layer integrity. The Bode plot confirmed the lower values of the resistance *R*_ct_ = 5872 × 10^2^ Ω cm^2^ and higher values of the capacity *Q* = 3465 × 10^−^ = F cm^2^ for sample WC–9FeNiCo–1Cr3C2–0.5NbC, which is evident from the appearance of the curves (as shown in Figure 10) where their slope remains constant. In contrast, the height of the sample plateau 1 at the frequency of 10 mHz increased. According to Figure 12, both samples’ phase angles were also high and close to 70°. The phase angle of the WC–9FeNi–1Cr_3_C_2_–0.5NbC sample was noticeably higher (70°) than that of sample WC–9FeNiCo–1Cr_3_C_2_–0.5NbC(65°) and the maximum was detected at medium frequency values (10 Hz).

### 3.3. Surface Analysis Results

For surface characterization purposes, five traces were equally distributed over the measurement surface of each sample. Parameters *R*a, *R*z, and *R*p were calculated based on filtered roughness profiles. The results are given as an arithmetic mean and range of the measured roughness parameters obtained for all traces, as shown in Table 6.

The presented results show no significant change in the surface texture after electrochemical corrosion. The analysis of the characteristic profiles of every measured trace also did not show considerable change due to the electrochemical corrosion process. There is no significant difference in the roughness profiles between the samples. An insignificantly more considerable change of parameters *R*z and *R*p was measured on sample 1 of the WC–9FeNiCo–1Cr_3_C_2_–0.5NbC composition. The traces on sample WC–9FeNiCo–1Cr_3_C_2_–0.5NbC before and after corrosion testing are shown as an example of a characteristic roughness profile in Figure 13.

The linear method based on the contact measurement of the sample profile on a specific track was used to characterize the surface roughness and not cover the entire sample surface. The surface recharacterization measurement was performed to confirm the results and cover the greater surface area where the dissolution/corrosion process could occur. The results in Table 7 also show no significant differences in the sample’s surfaces after corrosion; therefore, this implies no selective corrosion of the different phases in the surface layer occurred and the formation of corrosion products.

### 3.4. SEM Analysis of Samples before and after Corrosion 

Microstructural analysis conducted with an SEM electron microscope using a 10 kV electron beam showed no significant changes to the sample surfaces after corrosion, i.e., no increase in the surface porosity or occurrence of corrosion products at the surface was visible. Therefore, by comparing the micrographs obtained before and after the corrosion test, the results of other surface characterization techniques applied for surface characterization throughout this study have been confirmed, as shown in Figure 14.

### 3.5. Instrumented Indentation Results

The instrumented indentation test IIT results presented in Table 8 include the indentation plane strain modulus (*E*), indentation modulus (*E*_IT_), indentation hardness (*H*_IT_), Vickers instrumented hardness (*HV*_IT_), stiffness (*S*), indentation work (*W*_total_)and the ratio of elastic work *W*_elast_ in *W*_total_ (*n*_IT_).

The micromechanical properties testing before corrosion showed comparable *E*, *E*_IT_, *HV*_IT_, and *S* values of WC–9FeNi–1Cr_3_C_2_–0.5NbC and WC–9FeNiCo–1Cr_3_C_2_–0.5NbC with differences in the standard deviation range.

After exposure to a corrosive acid medium, changes in the micromechanical properties of both samples were observed. The differences in the sample’s behavior before and after exposure to an acidic environment are shown in the force-depth diagrams.

The decrease in *E*, *E*_IT_, *S,* and *W*_plast_ was measured for the WC–9FeNi–1Cr_3_C_2_–0.5NbC sample. The measured decline coincided with the increase in *W*_elast_ during indentation due to the sample’s lower stiffness and modulus of elasticity. Furthermore, the sample WC–9FeNi–1Cr_3_C_2_–0.5NbC exhibited higher HV values in the post-corrosion state, which indicates the hardening of the sample’s surface. The hardening might be attributed to the corrosion-induced formation of the hard protective Cr_2_O_3_ oxide phase (hardness of ~2800 HV) on the sample surface. More detailed research on the chemical changes on the surfaces of the samples is planned. The hardening of the sample’s surface was also confirmed by applying a smaller load of 500 mN to eliminate the measuring error. The same trend was established for all tested samples.

The repeatability of the measured mechanical properties, in terms of the standard deviation, was approximately the same for the samples WC–9FeNi–1Cr_3_C_2_–0.5NbC before and after corrosion. A graphical comparison of the force-depth curves for the samples WC–9FeNi–1Cr_3_C_2_–0.5NbC before and after corrosion is presented in Figure 15.

The force-depth curve is shifted to the left for the WC–9FeNi–1Cr_3_C_2_–0.5NbC sample after corrosion, which indicates a decrease in the total work, represented by the area under the curve. The slope of the loading curve is equal for both samples, before and after corrosion. An increase in the elastic work of approx. 5% in *W*_total_ was noted for the WC–9FeNi–1Cr_3_C_2_–0.5NbC sample after corrosion. The slope of the unloading curve decreased, resulting in a lower S. 

More significant changes in the micromechanical properties were noted for the WC–9FeNiCo–1Cr_3_C_2_–0.5NbC samples. A more prominent decrease in *E*, *E*_IT_, and significant hardness drop was noted. Unlike the WC–9FeNi–1Cr_3_C_2_–0.5NbC sample, where hardening of the sample’s surface appeared, a decrease of approx. 350 HV was measured for the WC–9FeNiCo–1Cr_3_C_2_–0.5NbC sample after corrosion. Significant scattering and non-uniformity of the mechanical properties across the test surface, presented by standard deviation, was caused by a corrosive acidic medium. The measured hardness values varied in the range of 1627 to 2307 HV. A graphical comparison of the force-depth curves for samples WC–9FeNiCo–1Cr_3_C_2_–0.5NbC before and after corrosion is presented in Figure 16.

Comparing the force-depth curves showed a significant increase in *W*_total_ and the entire curve shifted to the right with a uniform increase in both the elastic and plastic work. The slope of the loading curve was lower for the WC–9FeNiCo–1Cr_3_C_2_–0.5NbC sample after corrosion, resulting in a lower modulus of elasticity and hardness. In contrast, the slope of the unloading curve was parallel for both samples, resulting in a comparable S. Although no corrosion products were observed by the surface texture measurement, degradation of mechanical properties was evident, which most probably corresponded to the partial leaching of the Co phase in the FeNiCo binder.

## 4. Conclusions

The following conclusions can be drawn from the conducted research:Two innovative types of WC-based cemented carbides with alternative binders, FeNi and FeNiCo, with the addition of Cr_3_C_2_ and NbC as GGIs were developed using one cycle sinter-HIP process, showing excellent mechanical properties and a good corrosion resistance.The presence of Co in the FeNiCo binder significantly influenced the electrochemical corrosion resistance of the nanostructured cemented carbides in an acidic solution. The corrosion rate of the WC–9FeNiCo–1Cr_3_C_2_–0.5NbC sample was 2.5× higher compared to the WC–9FeNi–1Cr_3_C_2_–0.5NbC sample, which can be attributed to the lower Co corrosion resistance in acid solutions where Co dissolution occurs, as reported in several studies.A surface texture analysis using two different methods did not indicate significant changes in the surface texture, regardless of the binder type after corrosion techniques.The grazing incidence XRD measurements of the samples upon corrosion testing, given the shallow analytical depth, showed only traces of the possible corrosion products on their surface, which could indicate the presence of small amounts of the corrosion-formed Cr_2_O_3_ oxide phase on the surface of the samples corresponding to the rhombohedral R-3c unit cell, in accordance with the JCPDS entry no. 01-084-0315.Micromechanical properties testing before corrosion showed comparable *E*, *E*_IT_, *HV*_IT_, and *S* values of WC–9FeNi–1Cr_3_C_2_–0.5NbC and WC–9FeNiCo–1Cr_3_C_2_–0.5NbC with differences in the standard deviation range. After exposure to a corrosive acid medium, changes in the micromechanical properties of both samples were noted. The WC–9FeNi–1Cr_3_C_2_–0.5NbC sample exhibited higher HV values in the post-corrosion state, which indicates the hardening of the sample’s surface. To confirm this, further research is planned. A significant drop of hardness values (approx. 350 HV), a more prominent decrease in *E* and *E*_IT_, and non-uniformity of the micromechanical properties in the post-corrosion state was observed for the WC-9FeNiCo-1Cr_3_C_2_-0.5NbC, which most probably corresponds to the partial leaching of the Co phase in the FeNiCo binder.Overall, the study’s novel approach to exploring the impact of strong binder chemical composition on consolidated materials’ corrosive behavior and the introduction of an alternative binder system with improved corrosion resistance makes this research highly innovative and significant. The study’s findings open new avenues for research and development in the field of materials science and have the potential to drive innovation and improve the performance of materials in harsh environments.

## Figures and Tables

**Figure 1 nanomaterials-13-01407-f001:**
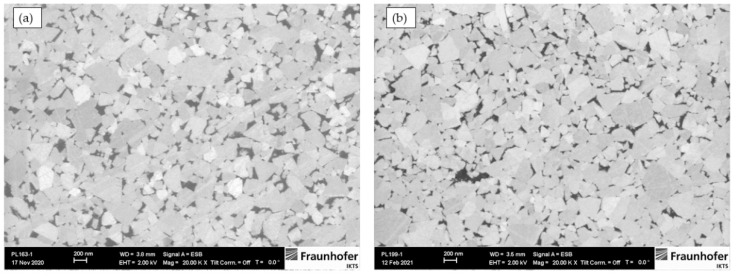
Microstructure of produced samples (**a**) WC–9FeNi–1Cr_3_C_2_–0.5NbC and (**b**) WC–9FeNiCo–1Cr_3_C_2_–0.5NbC.

**Figure 2 nanomaterials-13-01407-f002:**
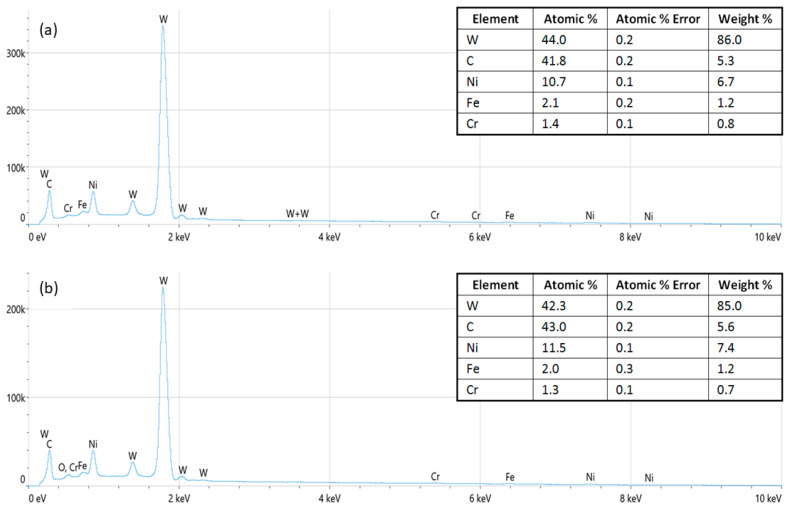
EDS spectra of the WC–9FeNi–1Cr_3_C_2_–0.5NbC sample (**a**) before and (**b**) after the corrosion process.

**Figure 3 nanomaterials-13-01407-f003:**
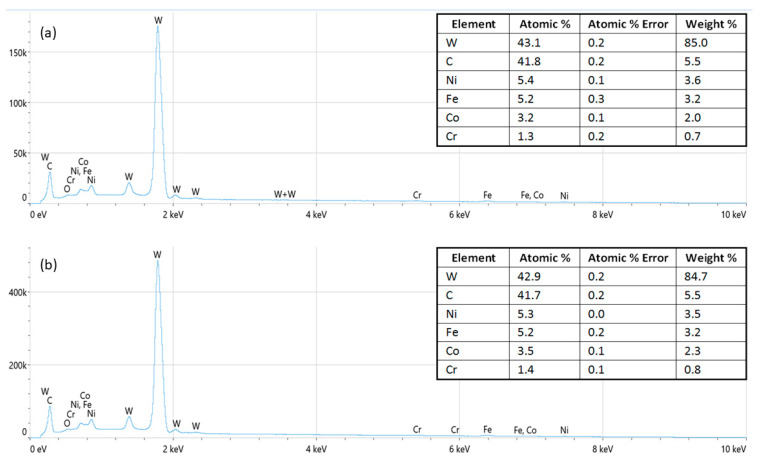
EDS spectra of the WC–9FeNiCo–1Cr_3_C_2_–0.5NbC sample (**a**) before and (**b**) after the corrosion process.

**Figure 4 nanomaterials-13-01407-f004:**
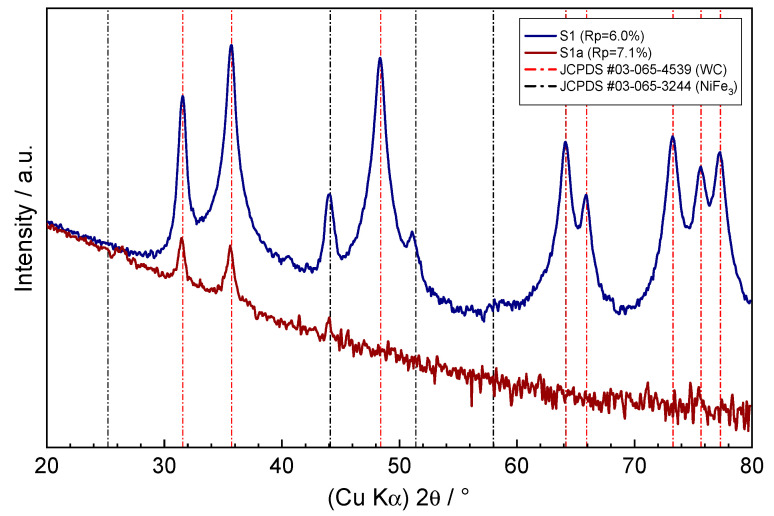
Grazing incidence XRD pattern of the WC–9FeNi–1Cr_3_C_2_–0.5NbC sample before (S1) and after the corrosion process (S1a).

**Figure 5 nanomaterials-13-01407-f005:**
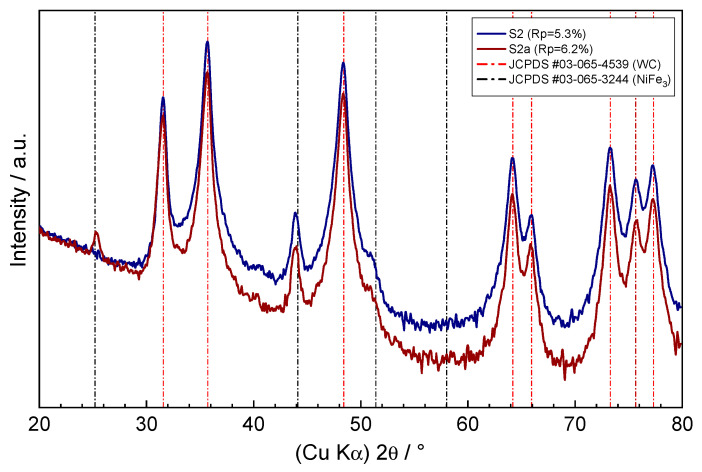
Grazing incidence XRD pattern of the WC–9FeNiCo–1Cr_3_C_2_–0.5NbC sample before (S2) and after the corrosion process (S2a).

**Figure 6 nanomaterials-13-01407-f006:**
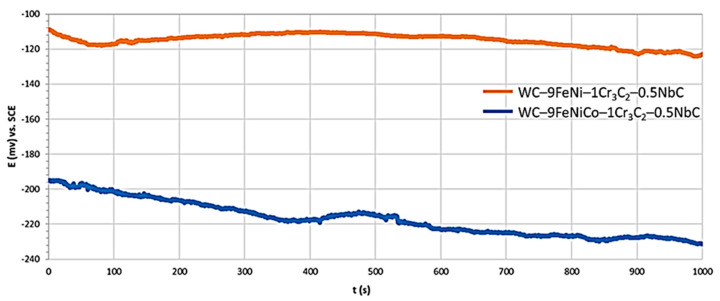
Open circuit potential (*E*_corr_) versus time curves of samples.

**Figure 7 nanomaterials-13-01407-f007:**
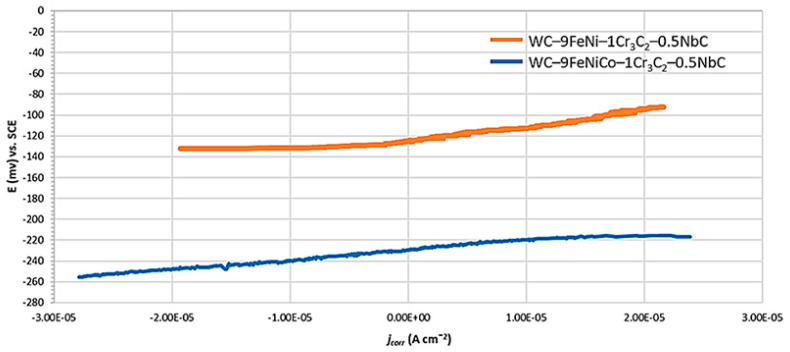
Linear polarization curves of samples.

**Figure 8 nanomaterials-13-01407-f008:**
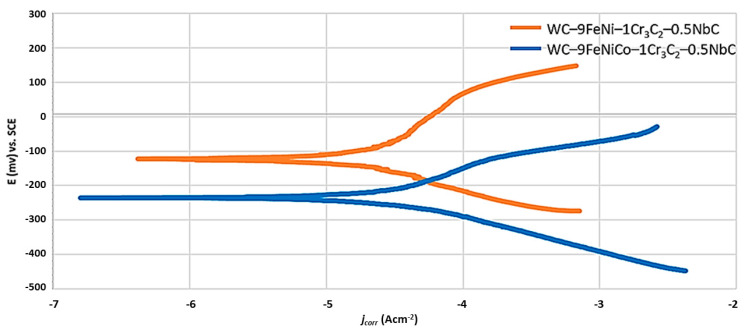
Potentiodynamic polarization curves (Tafel extrapolation diagram) of samples.

**Figure 9 nanomaterials-13-01407-f009:**
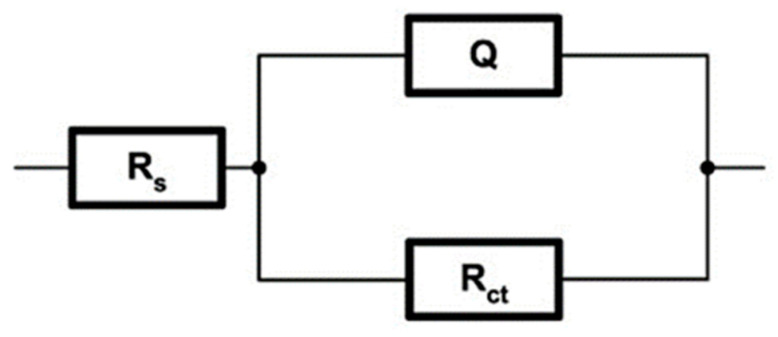
Model of R(QR) equivalent electrical circuit.

**Figure 10 nanomaterials-13-01407-f010:**
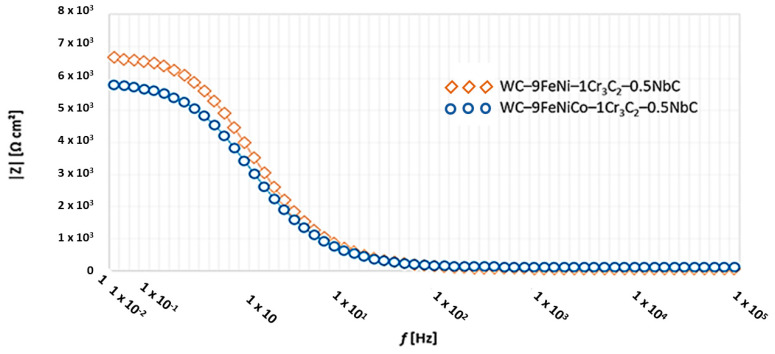
Bode plot of tested samples.

**Figure 11 nanomaterials-13-01407-f011:**
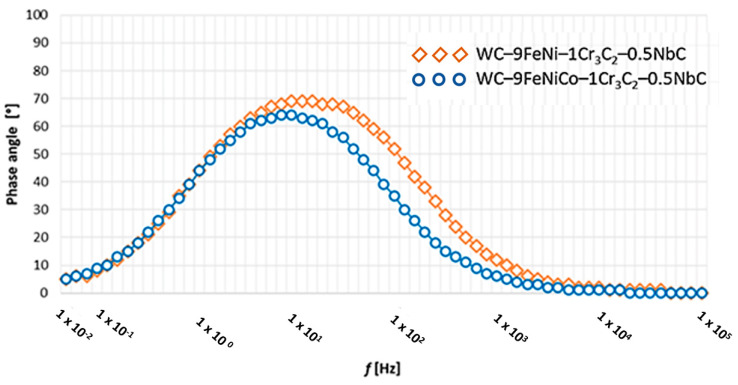
Nyquist plot of tested samples.

**Figure 12 nanomaterials-13-01407-f012:**
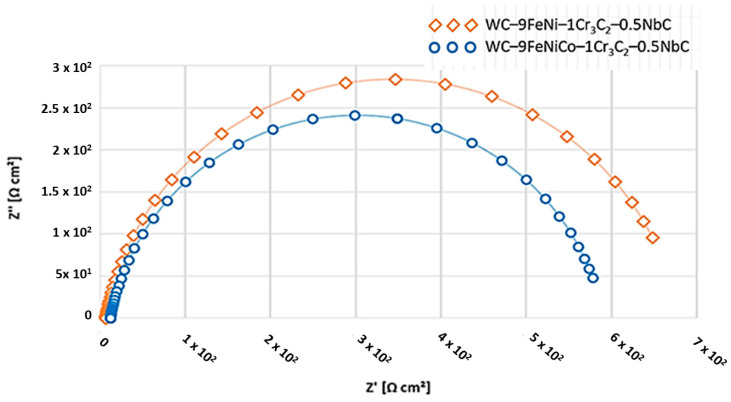
Phase angle as a function of the frequency of samples.

**Figure 13 nanomaterials-13-01407-f013:**
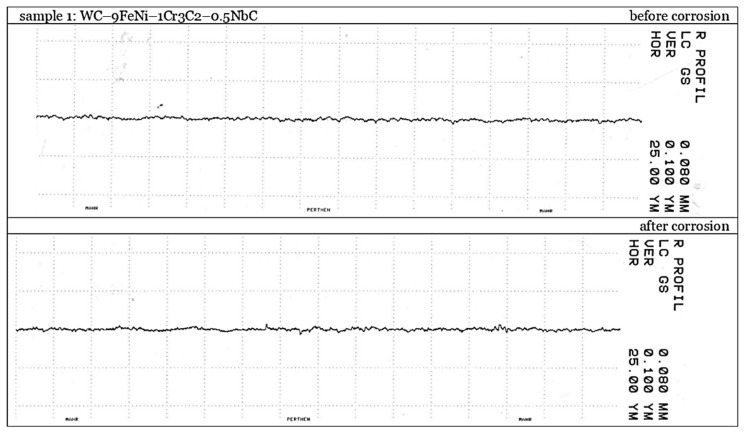
Characteristic profiles before and after electrochemical corrosion testing of sample 1: WC–9FeNi–1Cr_3_C_2_–0.5NbC.

**Figure 14 nanomaterials-13-01407-f014:**
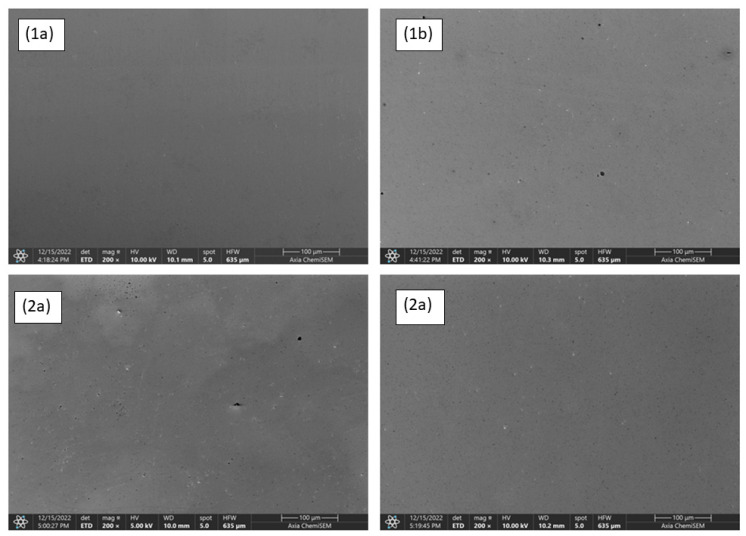
Microstructure of WC–9FeNi–1Cr_3_C_2_–0.5NbC sample (**1a**) before and (**1b**) after corrosion and sample WC–9FeNiCo–1Cr_3_C_2_–0.5NbC (**2a**) before and (**2b**) after corrosion.

**Figure 15 nanomaterials-13-01407-f015:**
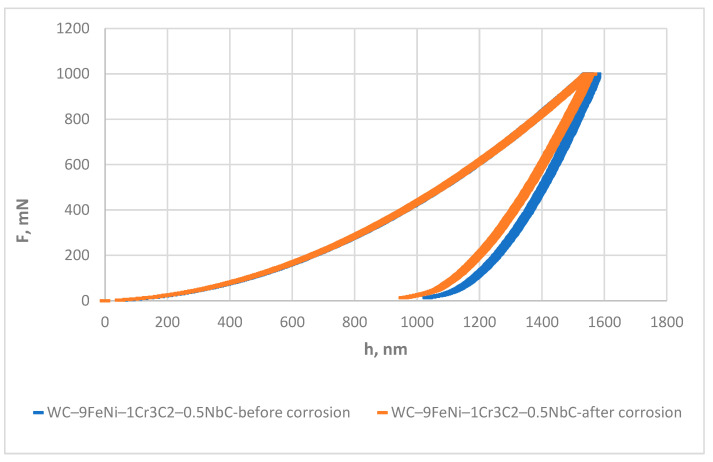
Force-depth diagrams of WC–9FeNi–1Cr_3_C_2_–0.5NbC samples before and after corrosion.

**Figure 16 nanomaterials-13-01407-f016:**
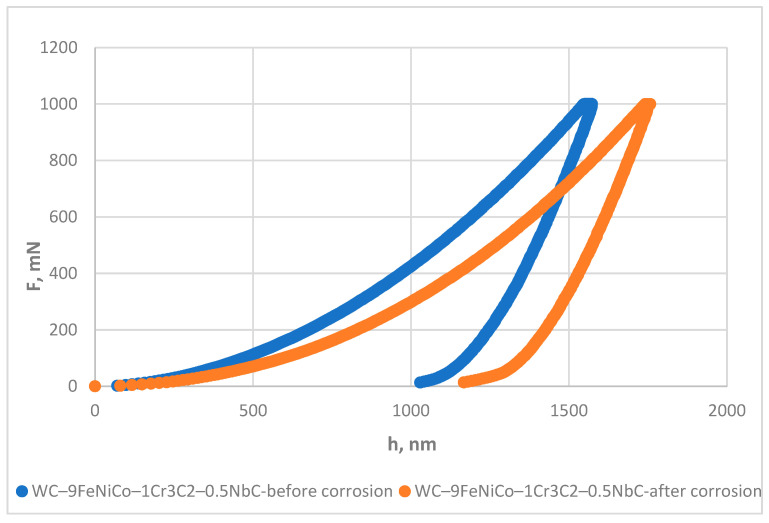
Force-depth diagrams of WC–9FeNiCo–1Cr_3_C_2_–0.5NbC samples before and after corrosion.

**Table 1 nanomaterials-13-01407-t001:** Starting mixtures characteristics.

Mixture	WC Powder	Grain Size *d*_BET_, nm	Specific Surface, m^2^/g	Binder Type	GGI, wt%
WC–9FeNi–1Cr_3_C_2_–0.5NbC	WC DN 3.0	<150	2.9	FeNi	1 wt% Cr_3_C_2_
WC–9FeNiCo–1Cr_3_C_2_–0.5NbC				FeNiCo	0.5 wt% NbC

**Table 2 nanomaterials-13-01407-t002:** Surface texture measurement parameters.

Texture Analysis Parameters Obtained by Perthometer S8P	
Temperature, °C	20 ± 0.3
Filtering	Gauss
Cut-off, mm	0.08
Tip radius, µm	5
Evaluation length, mm	0.4

**Table 3 nanomaterials-13-01407-t003:** Surface texture analysis parameters obtained by Alicona IF SensorR25: 50×.

Alicona IF SensorR25: 50×, Measurement Parameters	
Working distance, mm	10.1
Lateral measurement range (X and Y), mm	0.4
Lateral measurement area range (X × Y), mm^2^	0.16
Vertical resolution, nm	20
Min. measurable roughness *R*a, µm	0.08
Min. measurable roughness *S*a, µm	0.05

**Table 4 nanomaterials-13-01407-t004:** The results of electrochemical DC techniques.

Sample	*T*s [°C]	*E*_corr_ vs. SCE [mV]	*R*p [kΩ cm^2^]	*β*a [mV/dec]	*β*c [mV/dec]	*i*_corr_ [μA/cm^2^]	*v*_corr_ [mm/y]
WC–9FeNi–1Cr_3_C_2_–0.5NbC	20 ± 2	−122.989	1.462	111.595	98.372	13.514	0.174
WC–9FeNiCo–1Cr_3_C_2_–0.5NbC	20 ± 2	−231.257	0.986	170.065	116.22	32.794	0.441

**Table 5 nanomaterials-13-01407-t005:** The results of the EIS technique.

Sample	*T*s [°C]	*R*s [Ω cm^2^]	*Q* [F cm^2^]	*n* _1_	*R*_ct_ [Ω cm^2^]
WC–9FeNi–1Cr_3_C_2_–0.5NbC	20 ± 2	1.313 × 10^1^	2.868 × 10^−4^	0.891	6.736 × 10^2^
WC–9FeNiCo–1Cr_3_C_2_–0.5NbC	20 ± 2	1.186 × 10^1^	3.465 × 10^−4^	0.883	5.872 × 10^2^

**Table 6 nanomaterials-13-01407-t006:** Roughness measurement results before and after corrosion testing.

WC–9FeNi–1Cr_3_C_2_–0.5NbC	Sample 1						Sample 2					
	Before Corrosion			After Corrosion			Before Corrosion			After Corrosion		
	*Ra*	*Rz*	*Rp*	*Ra*	*Rz*	*Rp*	*Ra*	*Rz*	*Rp*	*Ra*	*Rz*	*Rp*
Arithmetic Mean, nm	2	16	9	2	18	12	2	16	9	2	17	10
Range, nm	0	4	2	0	6	4	0	6	4	1	4	2
WC–9FeNiCo–1Cr_3_C_2_–0.5NbC	Sample 1						Sample 2					
	Before Corrosion			After Corrosion			Before Corrosion			After Corrosion		
	*Ra*	*Rz*	*Rp*	*Ra*	*Rz*	*Rp*	*Ra*	*Rz*	*Rp*	*Ra*	*Rz*	*Rp*
Arithmetic Mean, nm	2	15	10	2	22	16	2	16	8	2	14	9
Range, nm	0	6	10	1	18	19	0	7	2	0	0	8

**Table 7 nanomaterials-13-01407-t007:** Re-characterization results of the sample surface changes after corrosion using an optical measurement sensor.

Arithmetic Mean of 4 Measurement Locations, nm	Before Electrochemical Corrosion		After Electrochemical Corrosion	
	*Ra*	*Sa*	*Ra*	*Sa*
WC–9FeNi–1Cr_3_C_2_–0.5NbC	26	49	35	86
WC–9FeNiCo–1Cr_3_C_2_–0.5NbC	27	87	23	46

**Table 8 nanomaterials-13-01407-t008:** Instrumented indentation test results for WC–9FeNi–1Cr_3_C_2_–0.5NbC and WC–9FeNiCo–1Cr_3_C_2_–0.5NbC.

WC–9FeNi–1Cr_3_C_2_–0.5NbC							
-before corrosion-							
	*E*, GPa	*E*_IT_, GPa	*H*_IT_, MPa	*HV*_IT_ (HV)	*S*, mN/nm	*W*_total_, pJ	*n_IT_* [%]
Min	652	624	22,901	2161	3.128	547,780	33.64
Max	723	691	24,298	2293	3.276	562,695	35.44
Mean	687	657	23,560	2224	3.194	55,7003	34.38
St dev	26	25	442	42	0.057	5002	0.538
-after corrosion-							
	*E*, GPa	*E*_IT_, GPa	*H*_IT_, MPa	*HV*_IT_ (HV)	*S*, mN/nm	*W*_total_, pJ	*n_IT_* [%]
Min	566	541	24,645	2326	2.686	522,561	38.239
Max	629	602	25,970	2451	2.952	549,522	42.08
Mean	600	573	25,306	2389	2.826	537,314	40.15
St dev	20	19	490	46	0.080	7877	1.244
WC–9FeNiCo–1Cr_3_C_2_–0.5NbC							
-before corrosion-							
	*E*, GPa	*E*_IT_, GPa	*H*_IT_, MPa	*HV*_IT_ (HV)	*S*, mN/nm	*W*_total_, pJ	*n_IT_* [%]
Min	642	614	22,754	2148	3.008	543,663	35.082
Max	705	674	24,468	2309	3.242	557,266	36.407
Mean	666	636	23,786	2245	3.119	549,294	35.97
St dev	21	20	586	55	0.081	4046	0.381
-after corrosion-							
	*E*, GPa	*E*_IT_, GPa	*H*_IT_, MPa	*HV*_IT_ (HV)	*S*, mN/nm	*W*_total_, pJ	*n_IT_* [%]
Min	488	466	17,233	1627	2.977	538,630	34.173
Max	656	627	24,437	2307	3.252	587,168	38.638
Mean	573	548	20,147	1902	3.076	559,631	35.74
St dev	43	41	1982	187	0.078	13,327	1.268

## Data Availability

The data presented in this study are available on request from the corresponding author.

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
