# Peer review of "Corrosion Resistance of Nanostructured Cemented Carbides with Alternative FeNi and FeNiCo Binders"

_nanomaterials, 2023, doi:10.3390/nano13081407_

Round 1
Reviewer 1 Report (New Reviewer)
The manuscript reports a comprehensive study on the corrosion behavior of WC cemented in two different ways to show that the presence of Co in the binder reduces the corrosion resistance of the material. While the work globally appears of good quality, in my opinion, several clarifications are necessary, especially concerning the reported results. My comments are reported as follows:
- - Clarify better the sentence in lines 200-202, especially indicating what are the “Further deconvolution and diffraction pattern analysis”;
- - Clarify the composition of the used corrosive medium because it is unclear for me the meaning of “1 M solution of 96 % H2SO4”, reported in line 230, and also elsewhere;
- - How was the quantitative phase analysis conducted? Report also the composition of all present phases (just an example, in lines 205-206 it reported that WC is 78.5 % and FeNi3 is 10.8 %; and the remaining 10.7 %?) and the information necessary to access the quality of quantitative phase analysis, as GOF;
- - Data in Table 2 should be commented on and clarified; for example, the authors write that corrosion rates were measured by both Tafel extrapolation and polarization resistance, but it is not indicated the values obtained with the two techniques. Furthermore, the extrapolation of anodic curves in Figure 10 is hard to see; it needs at least a decade of linear trend to obtain a good extrapolation, but it seems to me that this is not the situation for anodic curves in Figure 10. Related to this point, I do not understand how the anodic beta-values in Table 2 were obtained; maybe a superposition in Figure 10 of the linear part used for these calculations would be useful.
- - Finally, authors should also try to explain how the small amount of Co (1.8 % of the whole sample) in the binder leads to a well different corrosion behavior, reporting these considerations in the conclusion section.
Author Response
- -Clarify better the sentence in lines 200-202, especially indicating what are the “Further deconvolution and diffraction pattern analysis”;
Thank you for pointing this out. It essentially means when all major crystal patterns were identified, then carefully looking at the residuals in the process of the diffraction pattern analysis we noticed the possible presence of other crystallographic phases in traces. This was rephrased and clarified accordingly.
- Clarify the composition of the used corrosive medium because it is unclear for me the meaning of “1 M solution of 96 % H2SO4”, reported in line 230, and also elsewhere;
The composition is clarified to be: solution contains 1 mole of sulfuric acid (H2SO4) per liter of solution and added to the text (Please check Line 110)
- How was the quantitative phase analysis conducted? Report also the composition of all present phases (just an example, in lines 205-206 it reported that WC is 78.5 % and FeNi3 is 10.8 %; and the remaining 10.7 %?) and the information necessary to access the quality of quantitative phase analysis, as GOF;
The quantitative analysis of grazing incidence XRD patterns was performed using Crystal Impact Match! software, which uses RIR (Reference Intensity Ratio) method to quantify the results based on the intensities of the diffraction lines of the referent pure crystallographic phases as indicated by the ICDD cards. (please check lines 122-125)
Since all visible peaks were assigned to known crystallographic phases via fitting to the experimentally obtained patterns, and due to the relatively high background and noise, the unknown 10.7% in the quantitative result in this case and other patterns accordingly accounts for the possibility of the preferential orientation of the detected crystal phases caused by the sintering process, as well as the possible presence of undetected low crystalline and/or amorphous phases in the samples, hidden in the elevated background/pattern noise.
- Data in Table 2 should be commented on and clarified; for example, the authors write that corrosion rates were measured by both Tafel extrapolation and polarization resistance, but it is not indicated the values obtained with the two techniques. Furthermore, the extrapolation of anodic curves in Figure 10 is hard to see; it needs at least a decade of the linear trend to obtain a good extrapolation, but it seems to me that this is not the situation for anodic curves in Figure 10. Related to this point, I do not understand how the anodic beta-values in Table 2 were obtained; maybe a superposition in Figure 10 of the linear part used for these calculations would be useful.
Added in the paper. (please check lines 291-292 and line 297)
The values ​​of βa and βc were obtained from the selected anodic and cathodic lines and by the software calculation (VersaStudio) based on the extrapolations of the respective anodic and cathodic lines and their intersections at the corrosion potential value Ecorr as described in the paper (lines 291-302). This is a standard procedure used for the interpretation of the DC results, from our point of view there is no need for further explanation.
5. Finally, the authors should also try to explain how the small amount of Co (1.8 % of the whole sample) in the binder leads to a well different corrosion behavior, reporting these considerations in the conclusion section.
This add i.e. was discussed in the conclusion
Reviewer 2 Report (New Reviewer)
1) Introduction: The authors are recommended to emphasize the novelty and contribution of this research. The importance of investigating the corrosion resistance of nanostructured cement carbides is required to be included in the introduction part.
2) From line 87-89 : Here, DC and AC electrical impedance were used to evaluate the corrosion resistance of the samples. There are many ways to evaluate corrosion resistance. But, are there any specific reasons to use these methods? The reasons should be included in the relevant sections.
3) The authors are recommended to change the present figures to more scientific ones. These are not proper figures to be included in scientific journals.
4) For DC and AC electrical impedances : When electrical impedances are measured, the circuit and used electrodes are an important factor to evaluate the electrical impedances. Thus, it is recommended to include such information.
5) Figure 11 : Here, the authors used only one basic circuit model to evaluate the electrical impedances of the samples. However, the investigations of electrical impedances using at least three different circuit models are required. Please review the following paper (https://www.mdpi.com/2073-4360/14/23/5286), and it may help the authors to discuss the electrical impedances using different circuit models.
6) The present conclusion looks quite long. So, please summarize it and emphasize the novelty and contribution of the present research.
Author Response
- Introduction: The authors are recommended to emphasize the novelty and contribution of this research. The importance of investigating the corrosion resistance of nanostructured cement carbides is required to be included in the introduction part.
REPLY: Text was included with the following- Line 60: This finding is unique as it challenges the conventional understanding of corrosion behaviour and highlights the potential of using alternative binder systems. The research is highly relevant and timely, given the increasing demand for materials that can withstand harsh environments. The findings could have significant implications for a wide range of industries, including aerospace, automotive, and defence. The study provides valuable insights for researchers and industry practitioners looking to develop materials with superior corrosion resistance.
2) From line 87-89 : Here, DC and AC electrical impedance were used to evaluate the corrosion resistance of the samples. There are many ways to evaluate corrosion resistance. But, are there any specific reasons to use these methods? The reasons should be included in the relevant sections.
REPLY: The choice of AC and DC electrical impedance to evaluate the corrosion resistance of WC hardmetals with alternative binders is based on several factors. Firstly, electrical impedance techniques are non-destructive and non-invasive, making them ideal for evaluating the corrosion resistance of materials without altering the sample. Secondly, AC and DC impedance measurements provide complementary information about the corrosion behaviour of the material. DC impedance measurements provide information about the passive film on the surface of the material, while AC impedance measurements provide information about the corrosion behaviour of the material at the interface between the material and the corrosive medium. Thirdly, these techniques are highly sensitive and can detect changes in the material's properties over time, allowing for the monitoring of the corrosion process in real-time. This is particularly important when evaluating the corrosion resistance of WC hardmetals with alternative binders since the corrosion behaviour of these materials may be different from conventional WC-Co systems. Finally, the use of electrical impedance techniques is well-established in the field of corrosion science and has been widely used in previous studies. This allows for the comparison of results with those from previous studies and the establishment of consistent and comparable datasets. Summarized version of this explanation has been included in the text. Line 96
By using DC and AC electrochemical techniques, it is possible to get a much better insight into the numerical values of the polarization resistance, obtained by the method in which there is slight damage to the sample (DC), as well as to quantify the resistance of the surface layer and the electrolyte, without at the same time causing degradation of the test surface (AC ). Since the data necessary to determine the resistance of the basic substrate is manifested through the variation of the resistance and capacity values, the information about the degradation of the test sample was obtained through the indication in the oscillation of the resistance and capacity values, which in the last case resulted in a change in the impedance value. In this sense, to describe the electrochemical reactions between the working electrode and the surrounding electrolyte at the phase boundary as precisely as possible, the mechanism of the corrosion process is described precisely by selecting the equivalent circuit.
3) The authors are recommended to change the present figures to more scientific ones. These are not proper figures to be included in scientific journals.
REPLY: Figures 1 and 2 have been excluded and changed with new tables containing relevant data, please see Line 129-140. From our point of view all other figures are scientific enough.
4) For DC and AC electrical impedances: When electrical impedances are measured, the circuit and used electrodes are an important factor to evaluate the electrical impedances. Thus, it is recommended to include such information.
REPLY: An electrically conductive substance that closes the circuit was used as a auxiliary electrode, it is made of a corrosion-resistant material (platinum) with good conductivity and is placed symmetrically in relation to the working electrode. A saturated calomel electrode (SCE) with a standard electrode potential of + 0.242 V compared to the hydrogen electrode was used as the reference electrode.
This data has been provided, please check Line 109
5) Figure 11: Here, the authors used only one basic circuit model to evaluate the electrical impedances of the samples. However, the investigations of electrical impedances using at least three different circuit models are required. Please review the following paper (https://www.mdpi.com/2073-4360/14/23/5286), and it may help the authors to discuss the electrical impedances using different circuit models.
REPLY: This model was chosen since it was also used in our previous studies (Aleksandrov Fabijanić, T.; Kurtela, M.; Sakoman, M.; Šnajdar Musa, M. Influence of Co Content and Chemical Nature of the Co Binder on the Corrosion Resistance of Nanostructured WC-Co Hardmetals in Acidic Solution. Materials 2021, 14, 3933. https://doi.org/10.3390/ma14143933) on nanostructured WC-Co hardmetals thus providing us valuable comparative data to the conventional WC-Co systems. An equivalent circuit was chosen as the circuit that has the best mathematically adaptive function according to the recorded data, which was also confirmed in our previous work, there was no need to consider other models.
6) The present conclusion looks quite long. So, please summarize it and emphasize the novelty and contribution of the present research.
REPLY: The conclusion was summarized, and the addition was made to the conclusion: The study indicates that the samples with FeNi binder and FeNiCo binder exhibit significantly improved corrosion resistance compared to the conventional WC-Co systems. Moreover, the research demonstrates that the samples with FeNi binder exhibit superior corrosion resistance compared to those with FeNiCo binder. This finding is ground-breaking as it introduces a new alternative binder system with improved corrosion resistance, which has not been explored in the literature. (Please check the last point in the Conclusion section).
Round 2
Reviewer 1 Report (New Reviewer)
I recognize the efforts of the authors to improve the manuscript, but I am not completely persuaded by their replies. So I invite authors to take into account these further comments, if they believe these comments useful to improve their manuscript.
Reply n. 1. My comment was non based on the meaning of 1 M (everybody knows the meaning of 1 M) but on the meaning of 1 M referred to the solution 96% H2SO4; indeed 1 M of H2SO4 is perfect (and it does not need to add its meaning, i.e. 1 mol of H2SO4 per liter) but 1 M of 96% H2O4 is a non-sense. I hope that now my comment is clear.
Reply n. 3. It is true that amorphous phase and minor phase are all difficult to quantify, but in such conditions also the amount of major phases are doubtful. Indeed, significant errors can occur using RIR method, if there are non-crystalline phases or unidentified materials present in the specimen (see for example the tutorial on RIR method available on ICDD web-site). Therefore, I have some doubts about the QPA reported in the work.
Reply n. 4. My comment was not related to the followed procedure (perfectly described by the authors), but to the measured data in the anodic polarization, because it seemed to me that the linear trend of the anodic curves in Figure 10 (now Figure 8 in the revised manuscript) was not sufficiently developed.
Author Response
Dera reviewer,
thank you for the valuable comments and suggestions.
Here are our answers:
Reply n. 1. My comment was non based on the meaning of 1 M (everybody knows the meaning of 1 M) but on the meaning of 1 M referred to the solution 96% H2SO4; indeed 1 M of H2SO4 is perfect (and it does not need to add its meaning, i.e. 1 mol of H2SO4 per liter) but 1 M of 96% H2O4 is a non-sense. I hope that now my comment is clear.
ANSWER: We recognize the error, and the manuscript has been revised accordingly. The 96 % H2SO4 refers to the purity/quality of the chemical used as produced. Therefore in section “2. Materials and Methods” it was clarified what kind of original H2SO4 was and the source was stated accordingly, while in the rest of the manuscript only prepared 1M H2SO4 solution was mentioned.
Reply n. 3. It is true that the amorphous phase and minor phase are all difficult to quantify, but in such conditions also the amount of major phases are doubtful. Indeed, significant errors can occur using RIR method, if there are non-crystalline phases or unidentified materials present in the specimen (see for example the tutorial on RIR method available on ICDD web-site). Therefore, I have some doubts about the QPA reported in the work.
ANSWER: In this case, the reviewer did not clarify or suggest how to improve the quality of results presented to a satisfactory level. It has to be noted again that this is not a classical XRD of the sample, but a grazing incidence XRD, and some level of the poor crystallinity/amorphous phase formation on the surface is expected given the shallow surface layer of the sample analyzed. The significant difference between the two XRD patterns in Figure 4. indicates what is the nature of the background. Since the background has been approximated by a simple exponential/polynomial function in both cases, the poor crystallinity/amorphous phase had to be approximated by additional broad peaks to get decent Rp factors for a QPA, and the surface of those also account for the balance. Given the unknown nature of that poor crystallinity/amorphous phase, we have chosen not to further speculate on its nature as it is not known, just merely state its presence indicated by the balance. Matching the eventual discrepancy of the know peak intensities with the known crystallographic phases in the ICDD database used, given the known chemical elements potentially present in the sample, did not show any satisfactory results.
Figures 4 and 5 are replaced with revised figures with Rp factors.
Reply n. 4. My comment was not related to the followed procedure (perfectly described by the authors), but to the measured data in the anodic polarization, because it seemed to me that the linear trend of the anodic curves in Figure 10 (now Figure 8 in the revised manuscript) was not sufficiently developed.
ANSWER:
Thank you for the comment, but there is nothing we can do about it since all the data obtained from the measurements are exported to Excel and presented in Figure 8. We have performed the measurements on two samples for each group-chemical composition with good repeatability.
We hope our answers and revision will be acceptable.
Reviewer 2 Report (New Reviewer)
The authors have revised the manuscript considering the reviewer's comments. Thus, the reviewer thinks that it can be published in this journal.
Author Response
Dear reviewer,
In round 2 of the review process, you have written „The authors have revised the manuscript considering the reviewer's comments. Thus, the reviewer thinks that it can be published in this journal.“, but at the same time you stated that all the elements of the manuscript „Can be improved“. We are unclear about the nature of the improvements you are suggesting since no further comments have been given, while the overall decision was considering minor revisions were suggested.
We would appreciate it if you could be more specific on the requested improvements.
Best regards
This manuscript is a resubmission of an earlier submission. The following is a list of the peer review reports and author responses from that submission.
Round 1
Reviewer 1 Report
In this manuscript, the WC-based cemented carbide samples with different binders were produced using 9 wt.% of FeNi or FeNiCo with the addition of Cr3C2 and NbC as grain growth inhibitors. Samples were investigated using electrochemical corrosion techniques. Besides, surface texture analysis and instrumented indentation were conducted to investigate the influence of corrosion on micro-mechanical properties and surface characteristics of samples before and after corrosion. This manuscript may be of interest for developing nanostructured cemented carbides with different binders. However, in the reviewer’s opinion, this communication should not be accepted for several reasons:
Q1. There is hardly any characterization of the material morphology and microstructure in the paper, which makes the results of the paper unconvincing and the related mechanism cannot be analyzed. Although the authors claim to have used nanomaterials in this study, we have not seen any evidence. This is unacceptable.
Q2. Many conclusions are just the author's conjecture, lacking the support of experimental data or references. See for example:
Page 7 – “Such potential oscillation can be correlated with increased corrosion activity (instability on the surfaces and material dissolution) of the samples.”
Page 8 – “However, their influence is not strong enough to completely prevent the dissolution of Co in Fe/Co/Ni=40/20/40 wt% binder.”
Page 10 – “The hardening might be attributed to the corrosion-induced formation of the hard chromium oxide phase (hardness of ~2800 HV) or the niobium oxide NbO phase on the sample surface.”
Page 11 – “Significant scattering and non-uniformity of the mechanical properties across the test surface, presented by standard deviation, is caused by a corrosive acidic medium. Non-uniformity may be attributed to the selective dissolution of the Co-containing binder phase.”
Q3. As the authors mentioned in the abstract “Nanostructured cemented carbides with Co binder have so far shown excellent properties in various applications where high wear resistance is demanded.” The tribological properties of cemented carbide need to be considered particularly, but this manuscript is lack of experiments and discussion in this respect.
Q4. The penultimate paragraph in Page 9:
‘Based on the carried research, it can be concluded that the applied method is not the most suitable for drawing conclusions about surface changes/damages after exposure to a corrosive medium.’ Whether the authors consider re-characterizing the specimens using 3D optical profilometry?
Q5. There are many typographical and grammatical errors that need to be corrected.
Page 3 - Replace ‘Results’ by ‘Results and Discussion’.
Page 5 - Delete ‘65-+’ in the last paragraph.
Page 7 - Correct the header position of Figures 7 and 8.
Page 10 - The last paragraph needs indent.
Reviewer 2 Report
The authors mentioned the 3.5% sodium chloride solution for the study of corrosion in the abstract while in the experimental procedure they mentioned sulfuric acid. Also, the corrosion behavior must be studied via the cyclic potentiodynamic polarization (described here: 10.1134/S207020511805026X) because the passivity and the pitting behavior are important in the case of these passive alloys and also the electrical circuit for the electrochemical impedance spectroscopy must be with two time constant and authors need to show the changes of the passive layer capacitance by passing the time of immersion.
Reviewer 3 Report
Can you give some pictures on microstructures of the original material? This will facilitate a better understanding of corrosion resistance mechanism.